# Determination of Initial-Shear-Stress Impact on Ramsar-Sand Liquefaction Susceptibility through Monotonic Triaxial Testing

**Mehrdad Nategh [1], Abdullah Ekinci [2], Anoosheh Iravanian [3,*] and Siavash Salamatpoor [1]**

[1] Department of Civil Engineering, University College of Rouzbahan, Sari 3994548179, Iran;
nategh@rouzbahan.ac.ir (M.N.); ssalamatpoor@sci.iaun.ac.ir (S.S.)

[2] Civil Engineering Program, Middle East Technical University, Northern Cyprus Campus,
Kalkanli, Guzelyurt, North Cyprus, via Mersin 10, Turkey; ekincia@metu.edu.tr

[3] Civil Engineering Department, Near East University, Lefkosa 99138, Mersin 10, Turkey

* Correspondence: anoosheh.iravanian@neu.edu.tr; Tel.: +90-533-879-8518

**Abstract:** Liquefaction risk assessment is critical for the safety and economics of structures. As the soil strata of Ramsar area in north Iran is mostly composed of poorly graded clean sand and the ground water table is found at shallow depths, it is highly susceptible to liquefaction. In this study, a series of isotropic and anisotropic consolidated undrained triaxial tests were performed on reconstituted specimens of Ramsar sand to identify the liquefaction potential of the area. The specimens are consolidated isotropically to simulate the level ground condition, and anisotropically to simulate the soil condition on a slope and/or under a structure. The various states of soil behavior are studied by preparing specimens at different initial relative densities and applying different levels of effective stress. The critical state soil mechanics approach for identifying the liquefaction susceptibility is adopted and the observed phenomena are further explained in relation to the micro-mechanical behavior. As only four among the 27 conducted tests did not exhibit liquefactive behavior, Ramsar sand can be qualified as strongly susceptible to liquefaction. Furthermore, it is observed that the pore pressure ratio is a good indication of the liquefaction susceptibility.

**Keywords:** triaxial test; initial shear stress; pore water pressure ratio; liquefaction; Ramsar sand

## 1. Introduction

Soil liquefaction is a phenomenon in which soil loses its bearing capacity and behaves like a liquid. After Marcuson [1], Youd et al. [2] defined liquefaction as "the act or process of transforming any substance into a liquid". This phenomenon can damage a broad range of constructions such as dams, roads, and embankments. The study of liquefaction to develop susceptibility assessment approaches can enable the construction of safer structures, saving money and lives. The Niigata earthquake (1964) accelerated intense research on liquefaction. The evaluation of the liquefaction susceptibility of sandy soil in high seismic risk regions needs both laboratory and in situ experimental research. These tests usually need sophisticated devices to replicate the behavior of soil during an earthquake [3–6]. Castelli et al., [3] and Lentini et al., [5] in a soil liquefaction study carried for seismic retrofitting works performed different assessments like seismic dilatometer Marchetti tests (SDMTs), combined resonant column (RCT), cyclic loading torsional shear tests (CLTSTs), and undrained cyclic loading triaxial tests (CLTxTs) on isotropically consolidated samples [3–6]. Experimental work on static liquefaction was originally performed by Castro [7], and Castro and Poulos [8]. Their work on static liquefaction using a triaxial apparatus provided a better perception of the liquefaction mechanism and its controlling parameters. The application of initial shear stress during sample consolidation is a method for studying

the treatment of anisotropically consolidated samples. Jafarian et al. [9] observed that this method plays a significant role in stimulating the liquefaction susceptibility of the soil condition on a slope subjected to high shear strain or under a structure that can tolerate high shear stresses. Castro [7] performed several static tests on anisotropically and isotropically consolidated undrained samples and distinguished three different types of behavior for sands, as shown in Figure 1.

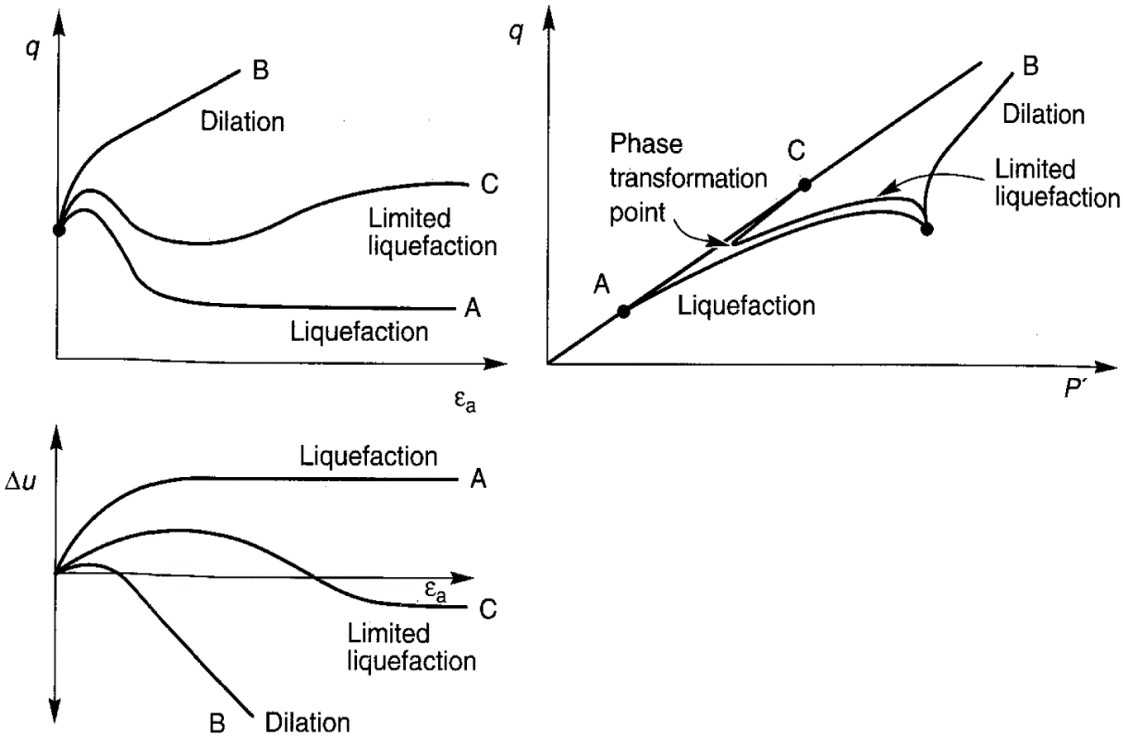

**Figure 1.** Liquefaction, limited liquefaction and dilation in static loading tests 7 [3].

On the other hand, a conventional triaxial test cannot represent a real site stress condition where the direction of the major principal stress may change according to various loading conditions such as wave load, traffic load or earthquake load. However, the mechanism of liquefaction is intensively related to these loading conditions. It has been widely reported in the literature that the rotation of the major principal stress orientation can accelerate the development of pore water pressure and can lead to liquefaction of sandy soils [10,11].

Numerous studies [8,12–18] have concluded that the initial state expressed in terms of the initial void ratio $e_0$ as well as the initial effective mean stress $p'_0$ significantly affect undrained soil response. The influence of the initial shear stress on the liquefaction susceptibility has been investigated in recent decades. Kramer and Seed [19] observed that in samples consolidated to principal effective stress ratios (K' of 1.5, 2.0, and 2.25, the increase in deviator stress under undrained conditions required to initiate liquefaction were approximately 0.6, 0.25, and 0.13 ksc (59, 25, and 13 kPa), respectively. They reported that the resistance to static liquefaction in these samples decreased significantly as the initial shear stress level increased. Later contributions by Harder and Boulanger [20], and Seed and Harder [21] demonstrated that the presence of the initial shear stress ratio ($\alpha$) improved liquefaction resistance at high relative densities (55–70%), whereas the effect was less pronounced at low relative densities (approximately 35%). According to recent research [22,23], triaxial tests performed at higher initial shear stress parameter values could create both improving and aggravating effects on the liquefaction susceptibility of loose or very dense sands, depending on the given range of the initial shear stress ratio ($\alpha$).

Soil samples under shearing up to large strains tend to reach a state of continuous deformation under constant shear (q) and normal stresses (p'), where such occurrence is known as the ultimate

steady-state line (SSL). Soil at the SSL exhibits a relationship between the ultimate values of the deviatoric stress and mean effective principal stress. Therefore, soil behavior can be predicted by expressing the state of the effective confining stress and defining the location of this point relative to the steady-state line. Castro and Poulos [24] observed that in addition to the steady-state line position being a unique soil property, the inclination of steady-state lines varies extensively, even for apparently similar soils. The steady-state concept and SSL were further described by other studies [1,7,24]. Poulos et al. [8] systematically measured the steady strength through stress-controlled Consolidated Undrained (CU) triaxial tests. On the other hand, Roscoe et al. [25] studied the yielding of soils and reported that when soil was subjected to shear distortion, it began to shear at constant volume at a certain critical pressure. Similarly, according to Schofield and Wroth [26], if a soil specimen is continuously distorted, it reaches steady state, i.e., flow failure. Although the critical state was initially developed on clayey soils, several studies have attempted to adapt this framework to granular materials [27,28]. However, Been et al. [29] stated that such an attempt is challenging due to the difficulties in determining the normal consolidation line for such soils. Coop [30] clarified that this phenomenon involved stress formation at the particle contact points, affecting the compression response of granular materials. However, the attempts of Ferreira and Bica [31], Coop [32], Ekinci et al. [18], and Rezaian at al. [33] for adopting this framework to granular soils was successful.

To understand the critical state concept of granular materials, the microscale behavior of sandy soils was investigated. Cavarretta et. al. [34] studied the micromechanical behavior of coarse-grained soils utilizing a new technology to compute the particle shape and surface roughness, for measuring the particle contact stiffness and interparticle friction to relate the nature of fundamental particle behavior with the traditional test results (triaxial and oedometer). Although a link was established between the roughness of the particle surface and interparticle friction, the influence of the particle shape was more noticeable. In a similar study by Senetakis et al. [35], repeated interparticle shear testing showed a small decrease in the friction angle, which can be because of asperity damage during initial shearing. Recently, Zhang et. al. [36] investigated sands with a variety of minerology using shape analysis, particle crush tests, and one-dimensional compression tests. Authors reported that particle mineralogy could be a major factor affecting the strength or compressibility, rather than the particle shape. Moreover, Zhao et. al. [37] investigated the effect of the initial density of specimens in one-dimensional compression to evaluate particle breakage. They stated that specimens prepared with high relative density had a lower probability of failure and failure modes that were less extensive compared to the low-density specimens. Moreover, it was reported that the effect of the initial density on the probability of particle survival reduced after significant breakage. Additionally, loose specimens exhibited higher compression due to particle failure leading to more fine generation by the further crushing of the existing fragments.

In support of the anisotropical soil condition on a slope and/ or under a structure, McDowell and Bolton [38] proved that compared to isotropic or $k_0$ conditions, shearing was more effective at breaking particles. Furthermore, Coop et. al. [39] investigated particle breakage by performing ring shear tests and observed that particle breakage continued up to very large strains along with volumetric compression, which was observed even for tests at moderate confining stresses. Accordingly, Chandler [40] stated that the observed critical state at the strain levels reached by triaxial equipment was due to the counteracting dilative strains because of particle rearrangement and compressive strains because of particle breakage.

Liquefaction could be induced by static loading or cyclic loading. Fort Peck Dam and Nerlerk Berm failures are good examples for static liquefaction. Cyclic loading may take different forms like storm load, ice load and machinery load, on which Jefferies and Been [41] have given a good historical summary. The most tremendous one is, undoubtedly, that caused by earthquake shaking [42] in terms of the intensity of loading and extensity of damage triggered. The static loading concerned in this study is primarily in relation with monotonic loading which occurs due to a sudden increase in pore water pressure and thus confining effective stress during an earthquake.

In this study, the effect of the initial shear stress ratio on the potential liquefaction susceptibility was investigated through monotonic testing. This study was the first to evaluate the liquefaction susceptibility by relating the pore pressure ratio ($r_u$) with the initial shear stress ratio ($\alpha$) and explain the liquefaction phenomena in relation to the particle breakage mechanism. Furthermore, the adopted anisotropic testing mimics realistic scenarios such as the simulation of the soil condition on a slope and/or under a structure. Compared to level ground conditions (with an initial shear stress ratio of $\alpha = 0$), only a few experimental studies on the monotonic behavior of sands in sloping ground conditions ($\alpha > 0$) are available. Moreover, specimen testing in undrained triaxial condition reduces the testing time. Therefore, this study proposes a method for liquefaction susceptibility determination which is less time consuming, and less expensive than other widespread methods just like dynamic triaxial testing method.

## 2. Materials and Methods

### 2.1. Geographical and Geological Background

Ramsar is located in the northern part of Iran. It is about 250 km north of the capital Tehran. Moreover, the Ramsar region is situated in the west part of Mazandaran province, borders the Caspian Sea to the north and the Alborz Mountains range to the south. The geographic coordinates of Ramsar are located between latitudes 36°32′00″ to 36°59′11″ N and longitudes 50°20′30″ to 50°47′12″ E. Figure 2 represents the sampling area of the study.

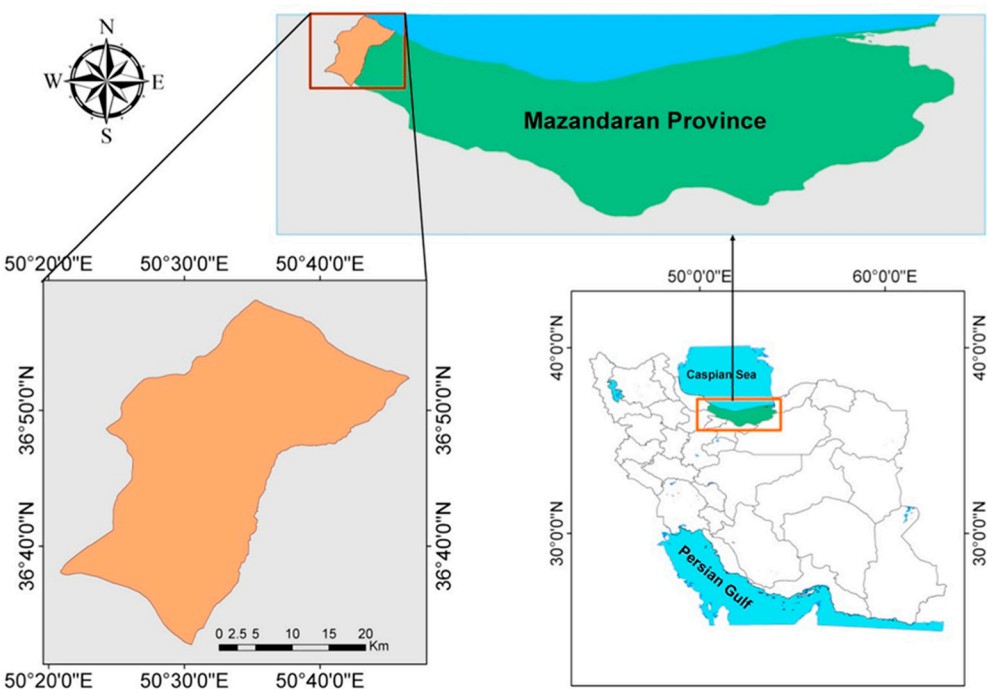

**Figure 2.** Geographical location of Ramsar area.

Geologically, this region is part of the Alborz–Azerbaijan zone, which is located in the northwestern part of the central Alborz Mountains, and except for a small part in the southwest corner, other outcrops are covered with forest trees. On the shores of the Caspian Sea, Quaternary deposits including alluvial plains and alluvial fans covered with farm fields and plants can be seen. In this area, a sedimentary sequence of river-flood facies with Quaternary marine sediments has formed. Tectonic movements are considered the most important factor in the growth and development of these sequences; especially the Middle Quaternary tectonic phase during which the northern Alborz Mountains were formed.

However, the history of these sediments is tied to the history of Alborz evolution, especially in the Quaternary [43]. The overall view of the geological structure of Ramsar can be seen in Figure 3.

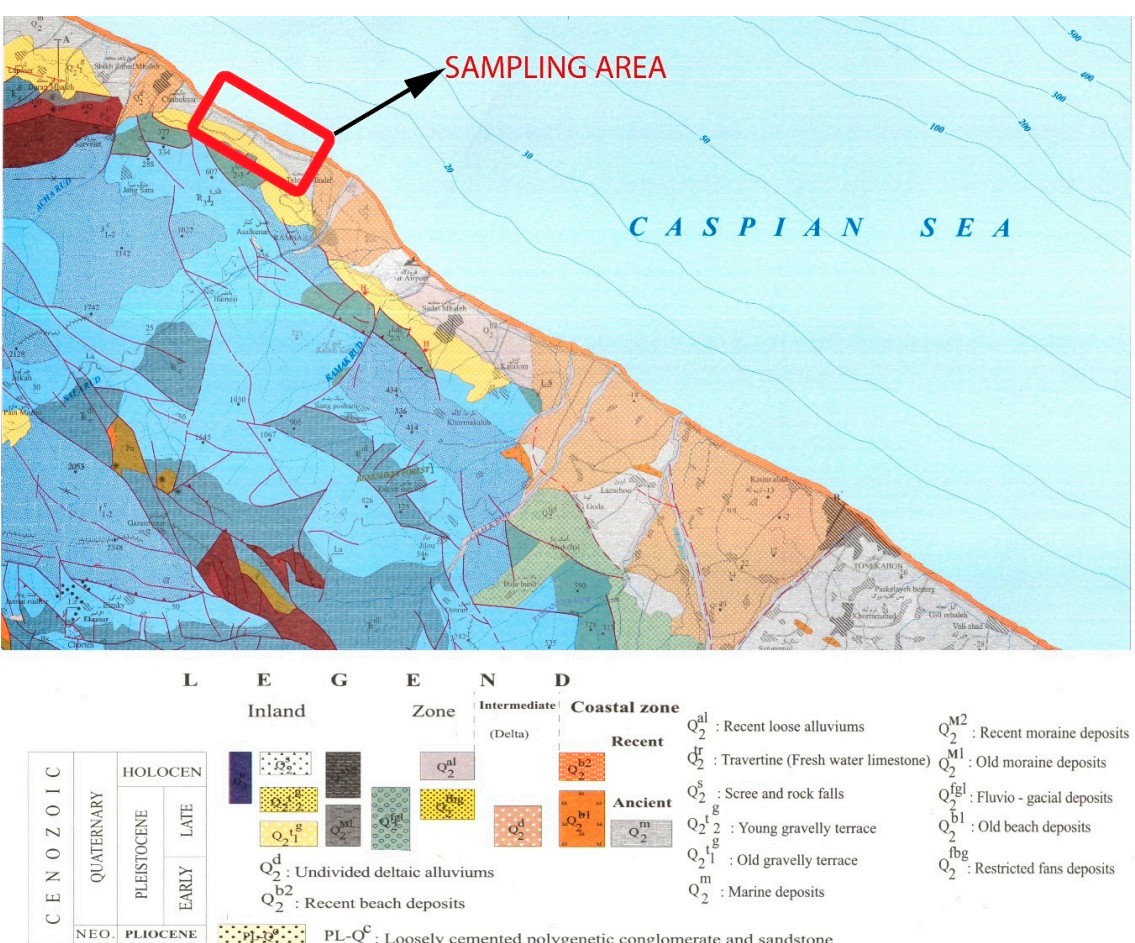

**Figure 3.** Geological and structural map of the sampling area (taken from 1:100,000 maps of Chalous, Shokran, Marzanabad, Baladeh and Ramsar, Geological Survey of Iran).

One of the most important factors in the geological and morphological structures of the region are faults. The existence of tectonic and geological activities has produced many faults in Mazandaran province, and almost more than 70% of the province and many urban and rural population areas are built up on the faults of the region [44].

### 2.2. Physical Parameters of Ramsar Sand

The relative density $D_r$ is a critical parameter that can control the stress–strain behavior or change the liquefaction susceptibility. In this study, three values of relative density (2%, 30%, and 45%) were considered to include very loose, loose, and medium dense sand, respectively. Ramsar sand is poorly graded clean sand which is extensively found in the southern coast of the Caspian Sea and classified as SP according to Unified Soil Classification System (USCS). The $e_{max}$ and $e_{min}$ values of Ramsar sand, measured according to ASTM D-4253 [45], and ASTM D-4254 [46], are 0.88 and 0.54, respectively. The mass of sand needed to prepare specimens with known relative densities was calculated and then it was gradually poured into the mold for each sample. The grain size distribution curve of Ramsar sand is depicted in Figure 4 and its basic properties are summarized in Table 1.

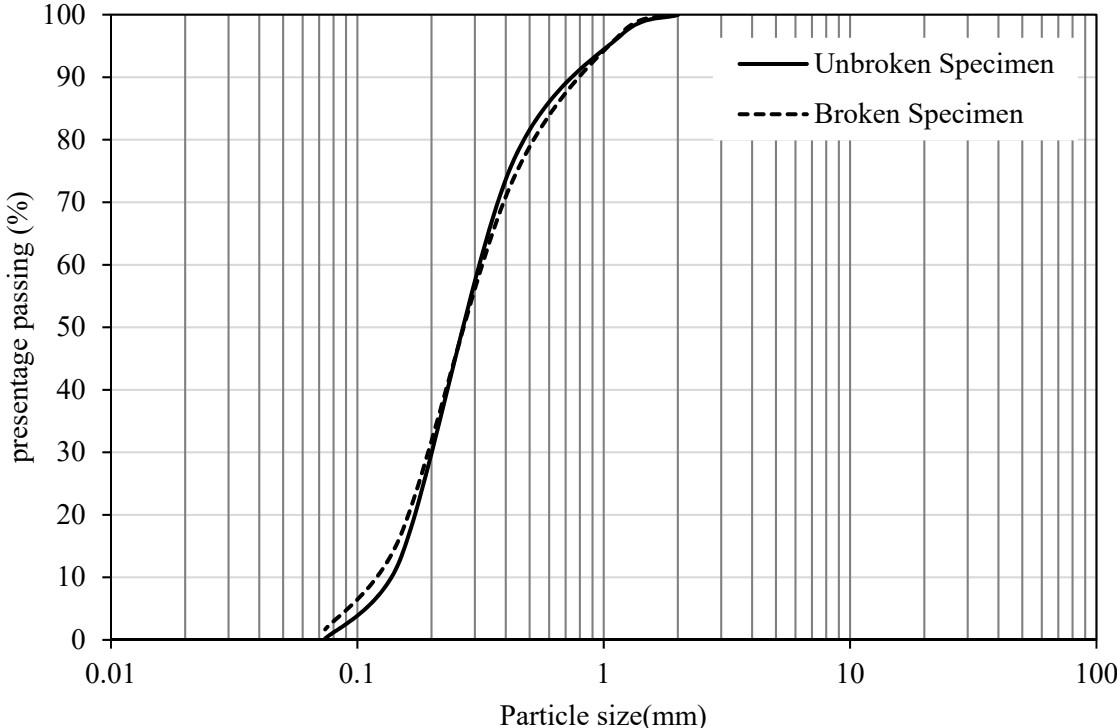

**Figure 4.** Grain size distribution before (unbroken) and after (broken) the performance of the triaxial test on Ramsar sand.

**Table 1.** Basic properties of intact Ramsar sand.

| Parameters | Values |
|---|---|
| Maximum void ratio, $e_{max}$ | 0.88 |
| Minimum void ratio, $e_{min}$ | 0.54 |
| Specific gravity, $G_s$ | 2.67 |
| $D_{50}$(mm) * | 0.22 |
| $D_{10}$(mm) * | 0.147 |
| $D_{30}$(mm) * | 0.187 |
| $D_{60}$(mm) * | 0.246 |
| Coefficient of curvature, $C_c$ | 0.96 |
| Coefficient of uniformity, $C_u$ | 1.67 |

* $D_{50}$, $D_{10}$, $D_{30}$, and $D_{60}$: the particle size diameters for which 50%, 10%, 30% and 60% of the sample is finer, respectively.

The grain size distribution of the specimens after triaxial testing was investigated and plotted against sand specimens obtained at 2 m depth from a borrow pit. Figure 4 shows that sand particles around 0.6–0.8 mm are further broken because there is a reduction in this particle percentage, whereas adversely, there is an increase in the particle percentage in the 0.1–0.2 mm particle size range. This observation reveals that after shearing, Ramsar sand particles are brittle. To identify the observed behavior in quantitative terms, the Hardin equation for relative breakage index ($B_r$) was adopted [47]. Hardin [47] defined breakage potential, $B_P$, as the area above the initial grading curve (which is the sand specimen before shearing in this case) up to the 100% passing line, confined between the lower-bound 74 μm minimum grain diameter and upper-bound 2 mm as the maximum grain diameter. Furthermore, total breakage, $B_t$, was calculated as the area between the particle size distribution curve of the initial grading curve and post crushed state (after shearing) grading curve. The breakage index is defined as the ratio of $B_t$ over $B_p$. For the grading curves seen in Figure 4, breakage potential ($B_P$) and total breakage ($B_t$) were found to be 60.3 and 0.74, respectively. As a result, the relative breakage

index ($B_r$) was found to be 0.012 and quantitatively confirmed the observed breakage of the Ramsar sand particles.

In order to further investigate this observation, sand particles before and after the test were examined using an optical microscope. The grains of the sand in Figure 5a are rounded with certain sharp ends, whereas in Figure 5b, the edges of the parent particle appear broken. Due to extensive shearing, more rounded particles are generated with very tiny spots related to broken bits of sand grains, clearly demonstrating the inherent brittleness. Moreover, the presence of silt is obvious in Figure 5.

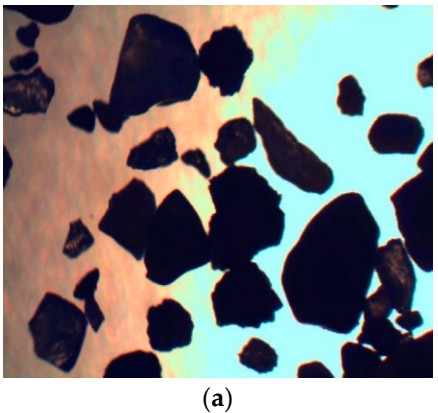 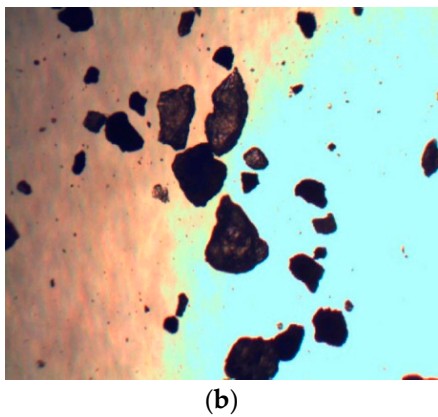

(**a**)                                     (**b**)

**Figure 5.** Microscopic view (40×) of Ramsar sand, where (**a**,**b**) are, respectively, before and after the test.

Furthermore, in order to assess the influence of the observed phenomenon during undrained shearing, two tests under the same confining pressure and relative density ($\sigma_{3c}$ = 350 kPa and $Dr_0$ = 30%) were conducted on broken and unbroken sand. In these two tests, the ultimate deviatoric stress ($q_u$) for broken sand was higher (283.85 kPa) than that for the unbroken one (255.9 kPa), indicating that the former has more resistance to contractive behavior. Similar to these findings, Cavarretta et. al. [34] had reported that compared to the as-supplied particles (perfectly rounded glass ballotini), crushed particles showed higher deviatoric stresses and more dilation.

### 2.3. Test Procedure

Moist tamping, which is the most common and valid method, was used to prepare the specimens. Sand specimens were oven-dried, mixed with 5% distilled water, and divided into five equal portions based on weight. A rubber membrane with a thickness of 0.3 mm was stretched by applying vacuum pressure. Each layer was then poured into a cylindrical split-mold, leveled with a spatula, and gently tamped. In order to improve the bonding between layers, the surface of each layer was scarified, and the same procedure was repeated for each layer. Measurement of the sample height and diameter was preceded by the application of a slight vacuum (no more than 20 kPa) on the specimen. The sample height and diameter were maintained at approximately 100 and 50 mm, respectively. Figure 6 illustrates a prepared sample. Subsequently, the cell was installed and filled with water. The vacuum was removed, and a positive pressure of approximately 30 kPa was applied. Carbon dioxide was infused through the specimen to augment saturation (to obtain a suitable degree of saturation, the carbon dioxide infusion technique proposed by Lade and Duncan [48] was used. Deaired water was then passed into soil under a specified back pressure to achieve a saturation of at least 95%. After consolidation, strain-controlled undrained loading (1 mm/min) was applied and continued until the occurrence of specimen failure or 30 mm axial displacement. Anisotropic tests were conducted with a very-low-rate axial load applied to the sample in a drained condition.

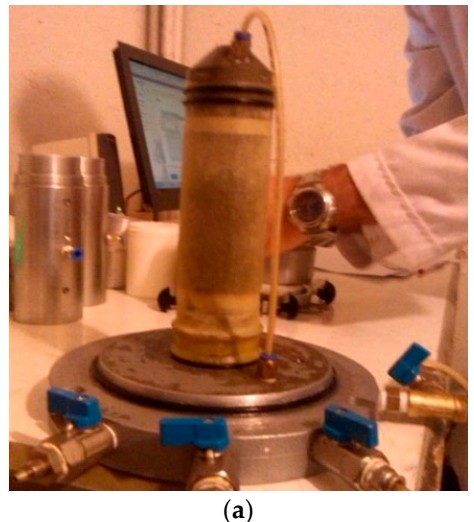

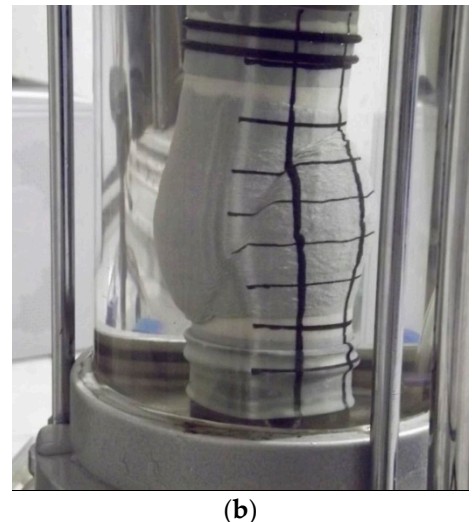

| (**a**) | (**b**) |

**Figure 6.** Examples of (**a**) preparation and (**b**) tested triaxial sample.

## 3. Results and Discussion

This study is primarily concerned with the effect of the initial shear stress on the potential susceptibility to the static liquefaction of Ramsar sand. It deals with specimens prepared at relative densities of 2%, 30%, and 45%. Isotropically (initial shear stress ratio of $\alpha = 0$) and anisotropically (initial shear stress ratio of $\alpha = 0.3$ and 0.5) consolidated specimens under three different confining pressures (150, 250 and 350 kPa) were tested in an undrained condition. A brief explanation of all 27 tests is presented in Table 2. In addition, all the stress path graphs are shown in Figure 7 (ACU and ICU denote anisotropically and isotropically consolidated undrained, respectively).

**Table 2.** Summary of the static triaxial tests conducted during the current study.

| # | Series | $\alpha$ | Situ | $\sigma_{3c}$ (kPa) | Dr0 (%) | Drc (%) | Result |
|---|--------|----------|------|---------------------|---------|---------|--------|
| 1 | $(R–A)_1$ | 0 | ICU | 150 | 2 | 12.1 | Liquefaction |
| 2 | $(R–A)_2$ | 0 | ICU | 150 | 30 | 37.5 | Liquefaction |
| 3 | $(R–A)_3$ | 0 | ICU | 150 | 45 | 51.5 | Dilation |
| 4 | $(R–A)_4$ | 0 | ICU | 250 | 2 | 16.6 | Liquefaction |
| 5 | $(R–A)_5$ | 0 | ICU | 250 | 30 | 42.7 | Liquefaction |
| 6 | $(R–A)_6$ | 0 | ICU | 250 | 45 | 53.77 | Dilation |
| 7 | $(R–A)_7$ | 0 | ICU | 350 | 2 | 22.48 | Liquefaction |
| 8 | $(R–A)_8$ | 0 | ICU | 350 | 30 | 37.37 | Liquefaction |
| 9 | $(R–A)_9$ | 0 | ICU | 350 | 45 | 55.35 | Limited |
| 10 | $(R–B)_1$ | 0.3 | ACU | 150 | 2 | 16.6 | Liquefaction |
| 11 | $(R–B)_2$ | 0.3 | ACU | 150 | 30 | 38 | Liquefaction |
| 12 | $(R–B)_3$ | 0.3 | ACU | 150 | 45 | 52.24 | Dilation |
| 13 | $(R–B)_4$ | 0.3 | ACU | 250 | 2 | 20.6 | Liquefaction |
| 14 | $(R–B)_5$ | 0.3 | ACU | 250 | 30 | 42.25 | Liquefaction |
| 15 | $(R–B)_6$ | 0.3 | ACU | 250 | 45 | 54.3 | Liquefaction |
| 16 | $(R–B)_7$ | 0.3 | ACU | 350 | 2 | 26.7 | Liquefaction |
| 17 | $(R–B)_8$ | 0.3 | ACU | 350 | 30 | 44.91 | Liquefaction |
| 18 | $(R–B)_9$ | 0.3 | ACU | 350 | 45 | 57.16 | Liquefaction |
| 19 | $(R–C)_1$ | 0.5 | ACU | 150 | 2 | 18.55 | Liquefaction |
| 20 | $(R–C)_2$ | 0.5 | ACU | 150 | 30 | 39.9 | Liquefaction |
| 21 | $(R–C)_3$ | 0.5 | ACU | 150 | 45 | 53.02 | Dilation |
| 22 | $(R–C)_4$ | 0.5 | ACU | 250 | 2 | 24.45 | Liquefaction |
| 23 | $(R–C)_5$ | 0.5 | ACU | 250 | 30 | 43.7 | Liquefaction |
| 24 | $(R–C)_6$ | 0.5 | ACU | 250 | 45 | 54.57 | Limited |

| # | Series | $\alpha$ | Situ | $\sigma_{3c}$ (kPa) | Dr0 (%) | Drc (%) | Result |
|----|----------|-----|------|---------------------|---------|---------|--------------|
| 25 | $(R–C)_7$ | 0.5 | ACU | 350 | 2 | 30.62 | Liquefaction |
| 26 | $(R–C)_8$ | 0.5 | ACU | 350 | 30 | 41.94 | Liquefaction |
| 27 | $(R–C)_9$ | 0.5 | ACU | 350 | 45 | 57.4 | Liquefaction |

ICU: Isotropic consolidated undrained triaxial tests, ACU: anisotropic consolidated undrained triaxial tests $\sigma_{3c}$: confining pressure, $D_{r0}$: molding relative density and $D_{rc}$: relative density after the consolidation.

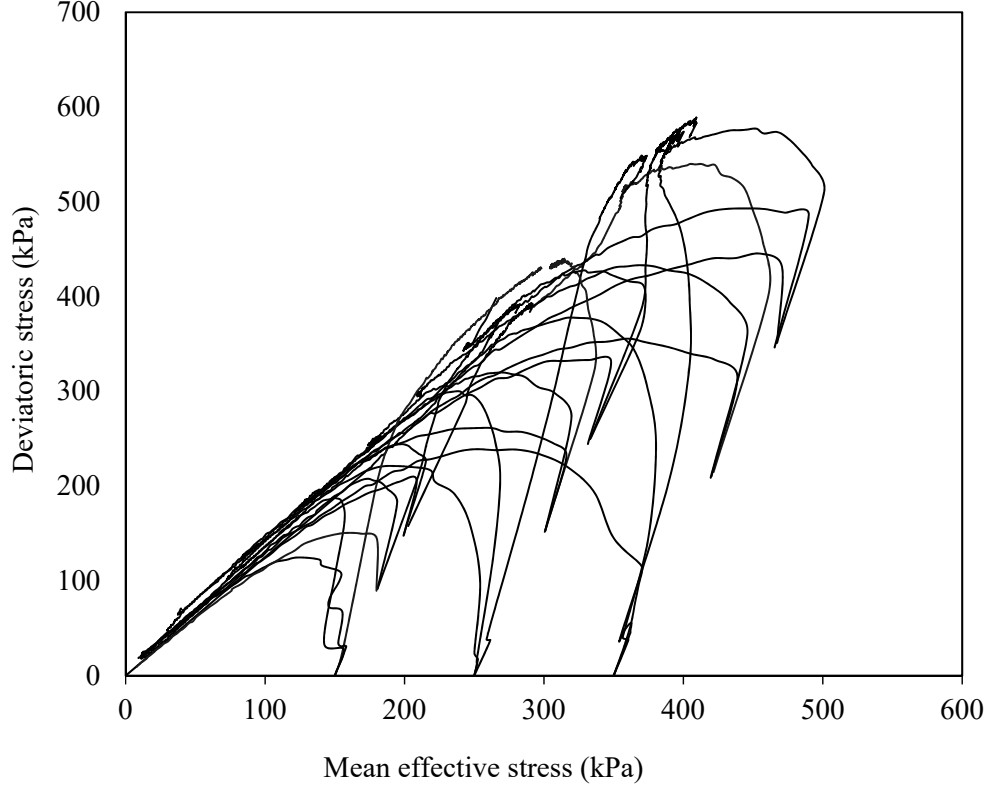

**Figure 7.** Steady state line in q–p′ plane with all 27 stress paths.

The SSL for the 27 CU tests of Ramsar sand are displayed in Figure 8. The brittleness of the particles causes sample breakage and fills in the available pores to bring the specimen to the same steady state as reported earlier by Cavarretta et al. [34]. Adversely, some researchers [18,49,50] had reported that poorly graded sand specimens prepared at varying relative densities had different SSLs. Ferreira and Bica [31], and Ekinci et al. [18] had tested a range of relative densities from minimum to maximum, and had managed to apply the critical state framework by grouping the relative densities and normalizing the results according to the group critical state line. Comparison of the relative densities of the specimens tested in this study with those of the study by Ekinci et al. [18] reveals that the densities of the specimens in this study falling into the Group which is located at a position closest to the NCL reach the same SSL.

### 3.1. Effect of Confining Pressure on Sand Behavior

In this study, sand behavior was controlled by the relative density and confining pressure, which, respectively, decrease and increase the liquefaction potential. Confining pressure is a parameter that relates to the depth and surcharge. Figure 9 displays the effective stress paths at different effective confining pressure at isotropic and anisotropic stress conditions for the initial relative density of the sand, $Dr_0 = 45\%$. It can be observed that upon increasing the effective confining pressure, the stress path tends to reach steady state, whereby there is a change in its direction, indicating liquefaction.

As stated by Rezaian et al. [33], the increase in confinement results in an increase in particle breakage, therefore since the particles are brittle, this means more breakage of particles and leads to reaching the critical state and results in liquefaction. Furthermore, Vaid et al. [51] and Bertalot et al. [52] stated that the increase in confining pressure results in an increase in liquefaction potential up to a limiting confinement. However, authors further mentioned that liquefaction is not solely dependent on the confining pressure, but also relates to the relative density of specimens and static shear stress level.

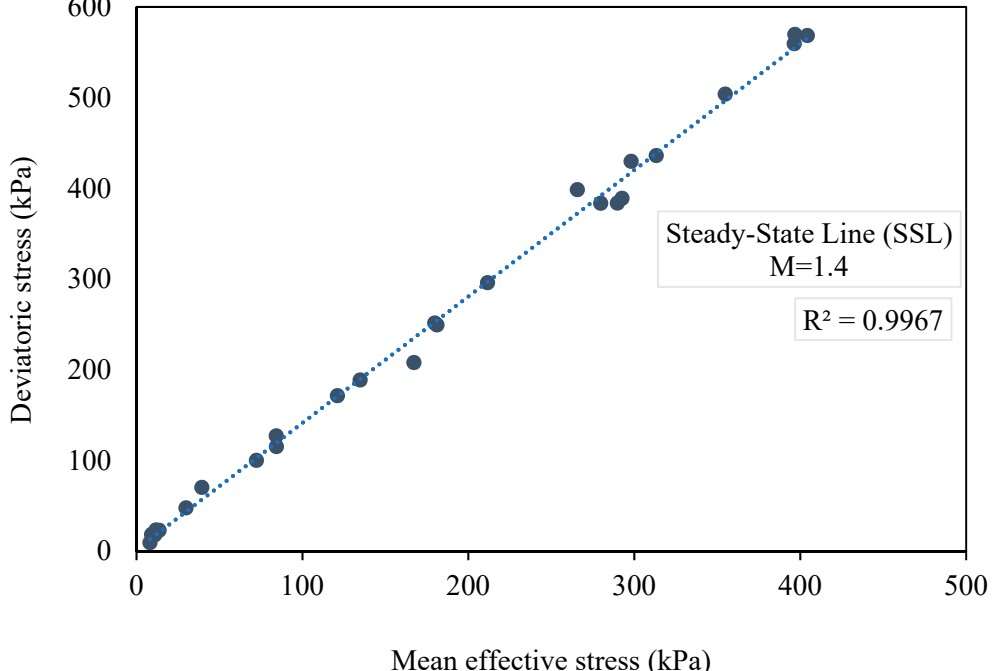

**Figure 8.** Steady state line in q–p′ plane.

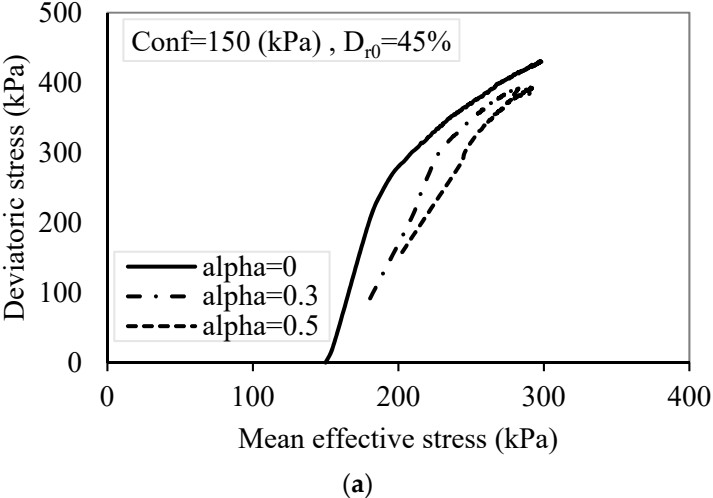

(**a**)

**Figure 9.** *Cont.*

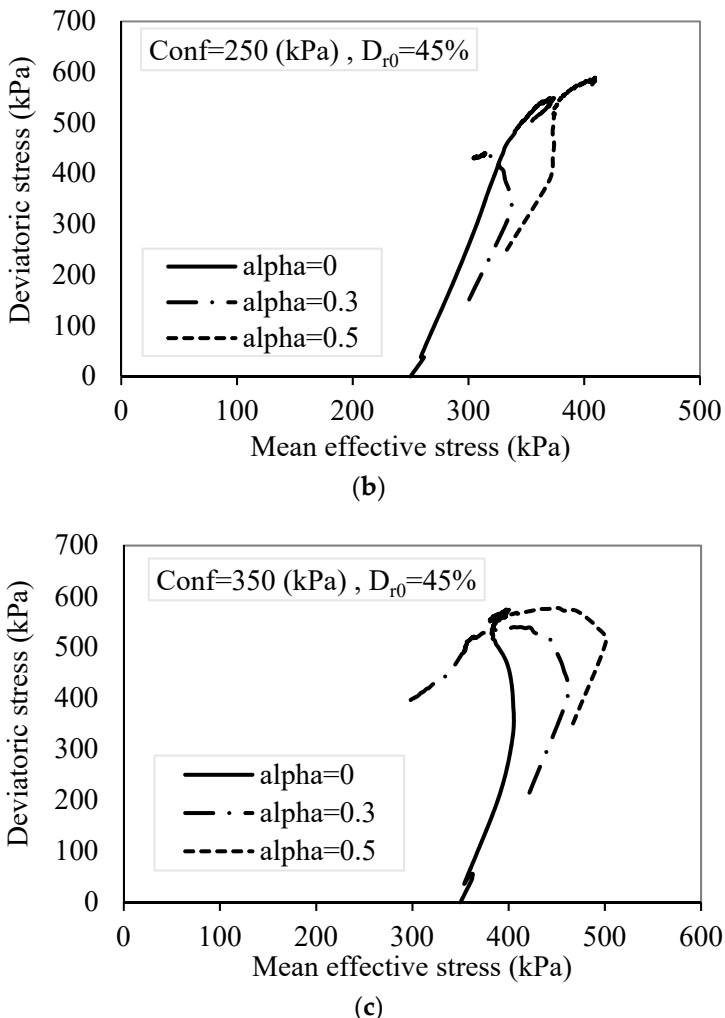

**Figure 9.** Effective stress paths at a different effective confining pressure: (**a**) 159 kPa; (**b**) 250 kPa; (**c**) 350 kPa in isotropic and anisotropic stress condition for $Dr_0 = 45\%$. Alpha ($\alpha$): the initial shear stress ratio.

### 3.2. Initial Shear Stress Ratio ($\alpha$)

The initial shear load parameter $\alpha$ obtained by anisotropically consolidating the specimens can differ according to certain soil parameters. Compared to level ground conditions with an initial shear stress ratio of $\alpha = 0$, only a few experimental studies on the monotonic behavior of sands in sloping ground conditions with higher initial shear stress ratios ($\alpha > 0$) are available, where the shear component is also present during the initial static condition. Therefore, this study (18 tests) includes anisotropically consolidated specimens prepared with initial axial loading in a drained condition. The initial shear stress ratio $\alpha$ is defined as the ratio of the initial axial stress ($q_s$) divided by twice the effective confining stress ($\sigma_{3c}$).

For example, when $\alpha = 0.5$ and $\sigma_{3c} = 150$ kPa, the load will be 0.29 kN, and for $\sigma_{3c} = 250$ kPa, the specimens will undergo a 0.49 kN load. Equation (1) indicates that the initial static shear stress level increases with the increase in $\alpha$ values, and the limiting case of $\alpha = 0$ represents level ground conditions without the initial static shear stress. Depending on the density of soil and the initial effective stress level, the presence of initial static shear stress may have either have an improving or aggravating impact on the resistance to liquefaction:

$$\alpha = \frac{\tau}{\sigma'_{3c}} = \frac{q_s}{2\sigma'_{3c}} = \frac{\sigma'_{1c} - \sigma'_{3c}}{\sigma'_{1c} + \sigma'_{3c}} \tag{1}$$

The excess pore water pressure (PWP) buildup of three specimens with a relative density of 45%, confining pressure of 150 kPa, and different initial shear stress ($\alpha$ = 0, 0.3, and 0.5) are shown in Figure 10. All the samples in these three tests exhibit dilative behaviors because of the high relative densities, whereas their PWP intensities differ depending on the initial shear stress ratio. In addition, upon increasing the initial shear stress, the PWP peak declines and subsequently, the intensity of the dilation increases. Due to the reduction in the excess pore pressure, the particle contact friction increases and causes resistance to liquefaction.

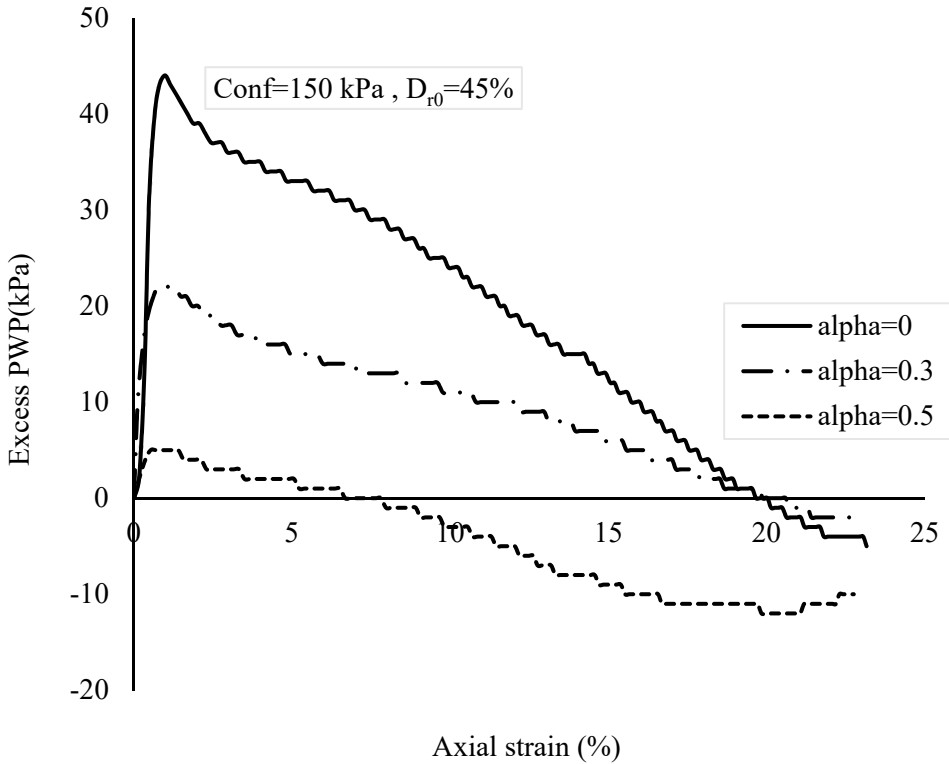

**Figure 10.** Comparative curvature for (R–A)$_3$, (R–B)$_3$ and (R–C)$_3$ in the u–$\varepsilon$% plane. Alpha ($\alpha$): the initial shear stress ratio.

Figure 11 displays different initial shear stress with higher confinement (250 kPa) compared to Figure 10. It can be seen that the initial shear stress not only changes the behavior intensity, but also the entire behavior to dilation in (R–A)$_6$, liquefaction in (R–B)$_6$ and limited liquefaction in (R–C)$_6$, where the initial shear stress ratios are 0, 0.3, and 0.5, respectively. This result is completely different from those observed in the specimens in Figure 10, where the excess PWP buildup is considerably more and has to be preserved up to high strain levels. The increase in PWP reduces the effective stress, which is the particle skeleton strength (reduction in the particle contact friction); therefore, such reduction leads to liquefaction. This may also show the inverse dependency of $\alpha$ on the effective confining pressure.

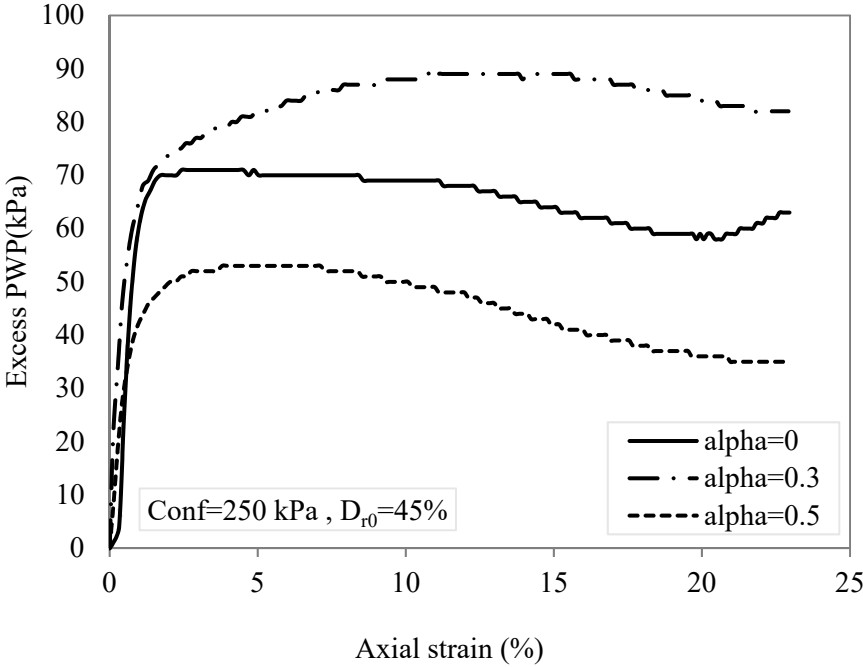

**Figure 11.** Comparative curvatures for (R–A)$_6$, (R–B)$_6$ and (R–C)$_6$ in u–$\varepsilon$% plane. Alpha ($\alpha$): the initial shear stress ratio.

### 3.3. Pore Water Pressure Ratio ($r_u$)

The observations in Figures 10 and 11 regarding the effect of confinement on the excess pore pressure, on varying the initial shear stress ratios, can be a criterion for liquefaction susceptibility. From Equation (2), it can be observed that the criterion for liquefaction susceptibility pore water pressure ratio ($r_u$) is the variation of the pore water pressure at failure to the initial effective confining pressure:

$$r_u = \frac{u_{excess\ at\ failure}}{\sigma'_{3c}} \tag{2}$$

The susceptibility ranges from 0–1, when $r_u$ approximates unity, and the liquefaction potential increases. Jafarian et al. [9] had also stated that it is common to have lower $r_u$ within the liquefaction phenomenon, termed as susceptible to liquefaction. Figure 12 presents the liquefaction susceptibility for isotropic and anisotropic specimens; the pore water pressure ratio ($r_u$) is plotted against the mean effective pressure at failure (Figure 12). It can be seen that most specimens are susceptible to liquefaction, while three of them show a negative value indicating that they have no liquefaction potential. In this study, the $r_u$ values range from −0.06–0.97. However, 21 out of the 24 tests show a positive value of $r_u$ indicating liquefaction behavior.

In Figure 12, the scattered pattern of points shows that $r_u$ is related to the relative density (D$_r$) because on increasing the effective confining pressure from 150 to 350 kPa, the dispersion decreases. The dispersion in tests with $\sigma_{3c}$ = 150 kPa reveals that except for one of the tests (R–C)$_2$, which may be an outlier, the outcomes of the other tests fall into two main opposing groups (phases): 1—highly susceptible and 2—completely non-susceptible to liquefaction.

As observed from the $r_u$ values plotted against the initial relative densities and from the point dispersion comparison shown in Figure 13, specimens with low D$_{r0}$ exhibit almost the same behavior, whereas for those with high D$_{r0}$, the tests results differ to a certain extent from each other. For medium D$_{r0}$, two distinguishable points, (R–A)$_2$ and (R–B)$_2$, indicate the commencement of dispersion. It is clear from Figure 13 that the increase in relative density reduces the liquefaction susceptibility. As stated by Zhao et al. [37], specimens with high relative density have lower particle failure probability and less extensive failure modes than loose specimens. Additionally, loose specimens exhibit higher

compression due to particle failure leading to more fine generation by the further crushing of existing fragments, and as stated earlier, reach the critical state, resulting in liquefaction.

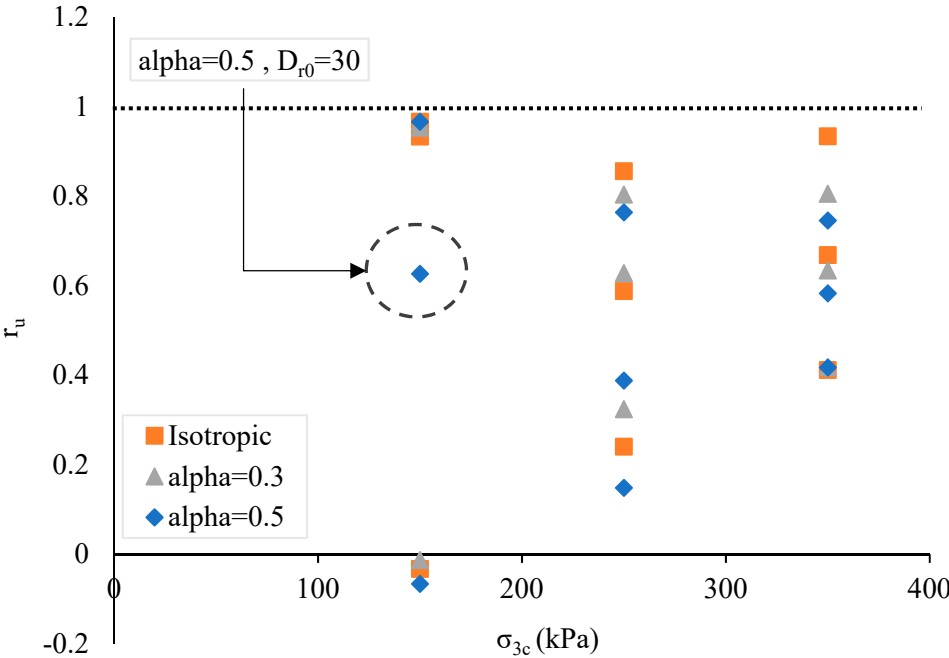

**Figure 12.** Pore water pressure ratio versus mean effective stress and effective confining pressure in isotropic and anisotropic stress condition. Alpha ($\alpha$): the initial shear stress ratio.

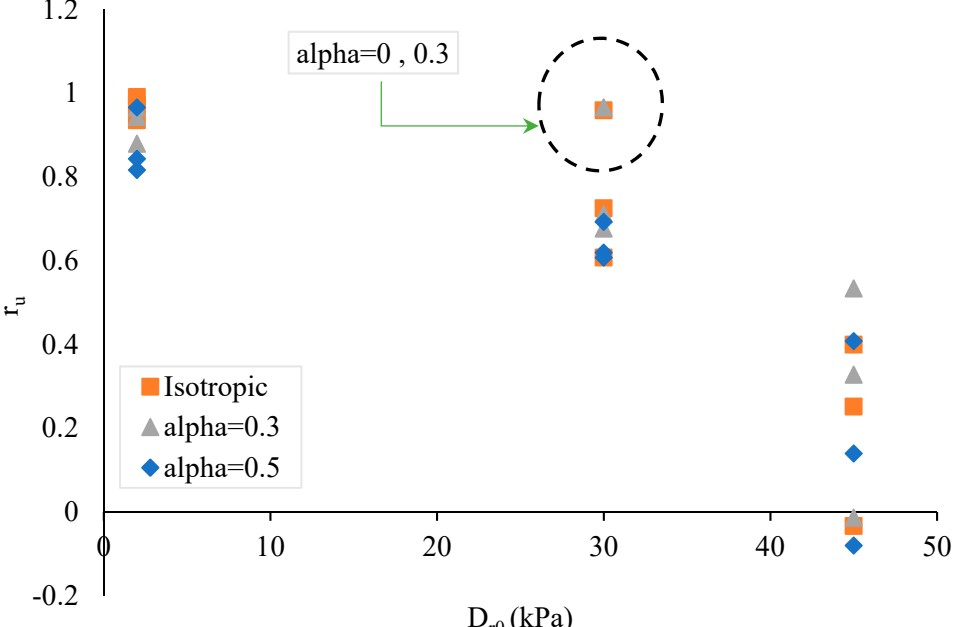

**Figure 13.** Pore water pressure ratio versus mean effective stress and effective confining pressure in isotropic and anisotropic stress conditions.

The friction angle at the failure point and $r_u$ have a negative correlation, as depicted in Figure 14. The friction angle ranges from 2° to 36° in this study, and $\phi'_{(deg)} = 2°$ and $\phi'_{(deg)} = 36°$ are related to specimens with high and low susceptibility to liquefaction, respectively. A low friction angle is

a result of excess pore water pressure, causing a loss of contact between particles which in turn results in interparticle friction loss.

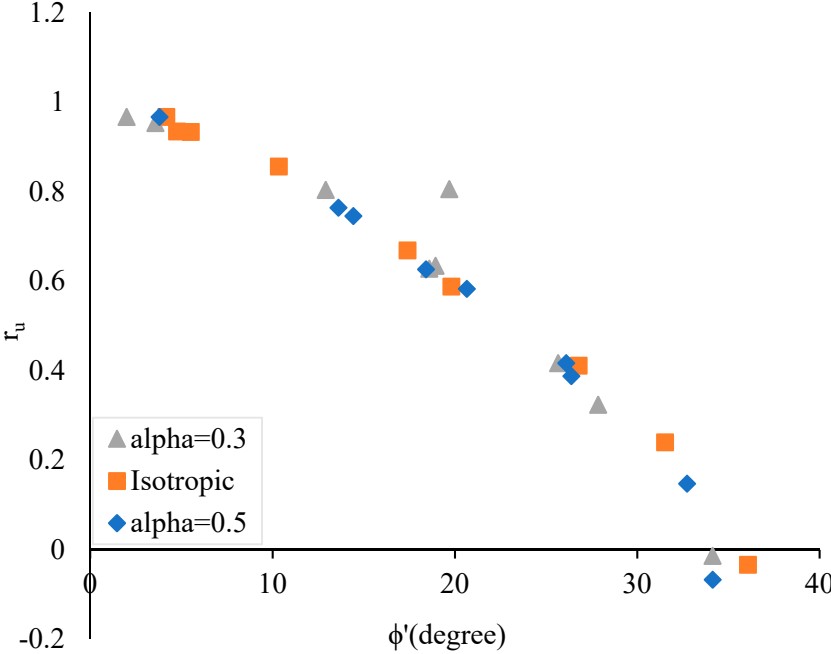

**Figure 14.** Pore water pressure ratio versus friction angle at failure in isotropic and anisotropic stress.

*3.4. Pore Water Pressure Ratio Versus the Initial Shear Stress Ratio*

Figure 15 displays the relationship between $r_u$ and $\alpha$ in three charts, whose effective confining pressures are different. It is obvious from the charts that on increasing the relative density, the liquefaction susceptibility reduces regardless of whether $\alpha$ and $\sigma_{3c}$ increase or decrease. This indicates that the relative density is much more effective than $\alpha$ and $\sigma_{3c}$. Figure 15a shows that the initial shear stress is not effective in loose sand (2%), whereas it influences semi-dense and dense specimens. In Figure 15b, this pattern slightly changes because the relative density after consolidation increases, which is strongly related to the effective confining pressure for loose sand. For example, as shown in Table 2, the relative density increases from 2% to approximately 23% after consolidation in sample $(R–A)_7$, whereas this growth is approximately 12% in sample $(R–A)_1$. This increment was calculated using the amount of water raised up during the volume change in the consolidation process. The specimen became denser due to the increase in $\sigma_{3c}$ from 150 to 250 kPa, and this pattern continued, when the effective confining pressure was increased from 250 to 350 kPa (Figure 15c). For the other relative densities, except for specimens with $\sigma_{3c}$ = 350 kPa, all the functions initially increased to $\alpha$ = 0.3 and then decreased to $\alpha$ = 0.5 which can indicate a lesser decreasing effect for higher values of $\alpha$ than for the medium values, for susceptibility under medium effective confining pressure. This phenomenon is also shown in Figure 12 by the change in test result from liquefaction to quasi-limited liquefaction, when $\alpha$ is increased from 0.3 to 0.5. In Figure 15c, when $\sigma_{3c}$ = 350 kPa, the pattern remains unchanged for $\alpha$ > 0. Moreover, the functions hardly change.

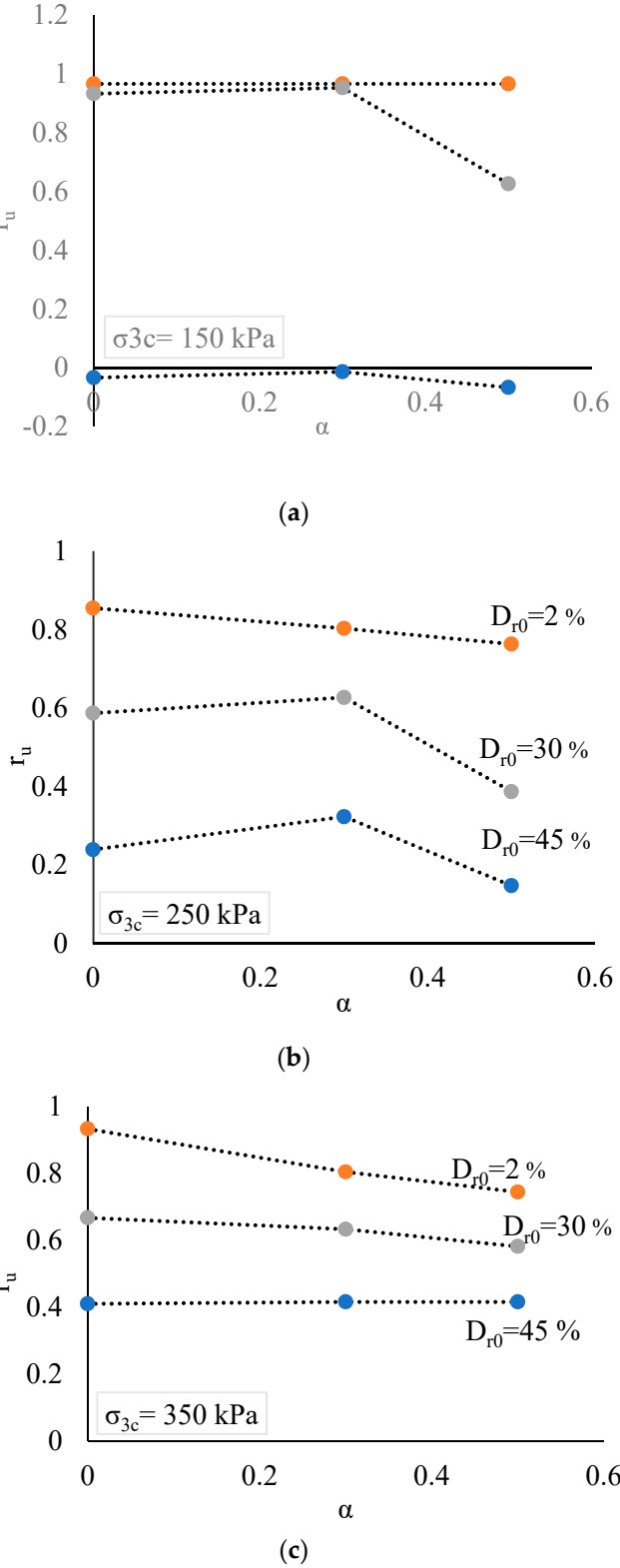

**Figure 15.** Pore water pressure ratio versus initial shear stress ratio at (**a**) 159 kPa, (**b**) 250 kPa and (**c**) 350 kPa effective confining pressure.

## 4. Conclusions

In this study, 27 isotropic and anisotropic monotonic undrained triaxial tests were performed to investigate the liquefaction susceptibility in relation to the critical state soil mechanics and

micromechanical behavior. It was observed that Ramsar sand can experience all the possible behaviors of liquefiable soils, namely, flow failure, limited liquefaction, and dilation. In addition, Ramsar sand is highly susceptible to liquefaction because 21 out of the 27 tests showed flow liquefaction. Among the other six tests, two showed limited liquefaction and only four showed dilation. Another important result was that the grains of Ramsar sand are brittle, which is responsible for its high susceptibility to liquefaction because a crushed specimen becomes more resistant to liquefaction ($q_u$-before = 255.9 kPa and $q_u$-after = 283.85 kPa). The initial shear stress $\alpha$ is a critical parameter for studying soil behavior on a slope and/or under a structure. Although it has a limited impact on loose sand, it can be an improving or aggravating factor depending on the effective confining pressure and relative density levels. The liquefaction susceptibility increased for an $\alpha$ range of 0–0.3 and reduced for a range of 0.3–0.5 under medium effective confining pressure and lower. However, at high effective confining pressure, $r_u$ values barely reduces as a result of increasing the initial shear stress ratio.

The pore water pressure ratio $r_u$ is a key parameter for evaluating sand behavior. In the conducted tests, even though only six tests did not result in liquefaction, only three of them showed negative $r_u$ values. For the other three tests with positive values of $r_u$, the susceptibility to liquefaction despite showing dilation or limited liquefaction is not surprising. Furthermore, a similar study is recommended to evaluate the findings of this study by investigating the varying particle minerology and percentage of fine generation due to particle breakage after shearing at various testing conditions.

Furthermore, it is recommended that a similar study be conducted by evaluating the findings of the proposed methods in light of scanning electron microscopy of sand grins before and after triaxial shearing to assess the effects of the brittle grains on liquefaction susceptibility.

**Author Contributions:** M.N. and S.S. conceived the study and were responsible for the scheduling and performing the experimental study. A.E. was responsible for structure interpretation. S.S. and A.I. wrote the first draft of the article and A.E. overlooked and finalized the study. All authors have read and agreed to the published version of the manuscript.

**Funding:** This research received no external funding.

**Acknowledgments:** The authors appreciate Kavosh Khak Azma Co. for providing triaxial apparatus and other laboratory equipment to conduct this study.

**Conflicts of Interest:** The authors declare no conflict of interest.

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
