# Peer review of "Determination of Initial-Shear-Stress Impact on Ramsar-Sand Liquefaction Susceptibility through Monotonic Triaxial Testing"

_applsci, doi:10.3390/app10217772_

Round 1
Reviewer 1 Report
General comment
The paper presents an experimental study on a poorly graded sand. Multiple topics are discussed but it is unclear what is the main subject of the work among the liquefaction, the breakage particles of the soils and the effect of the initial shear stress. In the current version, the aforementioned three subjects are mixed together but no one is investigated in an accurate manner. Unfortunately, the paper needs to be totally re-defined limiting the discussion to just one of three aspects. Personally, I think that the breakage of the grains of the Ramsar sand is the most original and interesting point to be studied and I encourage the Authors follow this pattern. In such a case, the study should be mainly focused on the soil properties including a wide description of site location, geological background, laboratory investigations aiming to assess the effects of the brittle grains (grain size distribution before and after, SEM and so on).
Minor comments
-The parameter a is not defined (row 63)
-Recurrent typos should be checked throughout the all manuscript, such as “some 1. Introduction” (row 33)
Reviewer 2 Report
Please see the attached file.

Reviewer 3 Report
This paper present a series of undrained triaxial test on the sandy specimen subjected to static shear. A total of 27 tests have been carried out for different confining pressure, different CSR and different initial conditions with or without considering the anisotropic consolidation of the soil. Overall, the paper is well written. However, the following points must be addressed before the manuscript can be published in this journal.
- Line 199 brakeage should be breakage
- This reviewer seldom see Dr=2% in the triaxial test. Would it be too small to represent a real site condition.
- Line 216. Theoretically speaking, liquefaction is more likely to occur in shallow soil layer with less significant surcharge and confining pressure. However, this paper show contradictory results. Can the authors explain something on this? Moreover, the breakage index should also be given to quantify the effect of particle breakage on the liquefaction susceptibility of the soil.
- Line 239. It is recommended that the authors use the liquefiable case rather than non-liquefiable case to compare the pore water pressure response with and without considering the initial shear.
- It is commonly known that conventional triaxial test such as those considered in this study can not represent a real site stress condition where direction of the major principal stress may change accordingly to various kinds of engineering activities such as wave load, traffic load or earthquakes. Indeed, the mechanism of liquefaction is intensively related to these engineering activities. It has been widely reported in the literature that the rotation of the major principal stress orientation can accelerate the development of pore water pressure and the liquefaction susceptibility of sandy soil (Towhata and Ishihara, 1985; Zhao et al. 2020). I recommend the authors to address this limitation in the paper by citing the suggested references where relevant.
References:
- Towhata, and K. Ishihara, Undrained strength of sand undergoing cyclic rotation of principal stress axes, Soils and Foundations 25(2) (1985) 135-147
- Y. Zhao and J. F. Zhu and J. H. Zheng and J. S. Zhang, Numerical modelling of the fluid–seabed-structure interactions considering the impact of principal stress axes rotations, Soil Dynamics and Earthquake Engineering, 136 (2020) 106242
Round 2
Reviewer 3 Report
The authors have made revisions to respond to the comments, and it is in a format suitable for publication.